# Breaking through Deterministic Barriers:
# Randomized Pruning Mask Generation and Selection

**Jianwei Li[1]**  **Weizhi Gao[1]**  **Qi Lei[2]**  **Dongkuan Xu[1]**

[1]North Carolina State University, {jli265, wgao23, dxu27}@ncsu.edu
[2]New York University, ql518@nyu.edu

## Abstract

It is widely acknowledged that large and sparse models have higher accuracy than small and dense models under the same model size constraints. This motivates us to train a large model and then remove its redundant neurons or weights by pruning. Most existing works pruned the networks in a deterministic way, the performance of which solely depends on a single pruning criterion and thus lacks variety. Instead, in this paper, we propose a model pruning strategy that first generates several pruning masks in a designed random way. Subsequently, along with an effective mask-selection rule, the optimal mask is chosen from the pool of mask candidates. To further enhance efficiency, we introduce an early mask evaluation strategy, mitigating the overhead associated with training multiple masks. Our extensive experiments demonstrate that this approach achieves state-of-the-art performance across eight datasets from GLUE, particularly excelling at high levels of sparsity.

## 1 Introduction

One of the main challenges in deploying large neural networks (such as BERT (Devlin et al., 2019) and GPT-3 (Brown et al., 2020)) in production is the huge memory footprint and computational costs. Meanwhile, studies show that large and sparse models often yield higher accuracy than small but dense models (Gomez et al., 2019). As a result, pruning has been popularized to dramatically reduce memory size and computational power consumption with little to no performance degradation. (Hoefler et al., 2021; Glorot et al., 2011; Kaplan et al., 2020; Li et al., 2020; Mhaskar and Poggio, 2016; Brutzkus et al., 2017; Du et al., 2018).

Pruning aims to eliminate redundant weights, neurons, and even layers in models. Many works focus on magnitude-based pruning (Hagiwara, 1993; Gale et al., 2019; Thimm and Fiesler; Han et al., 2015; Zhu and Gupta, 2017; Cuadros et al.,

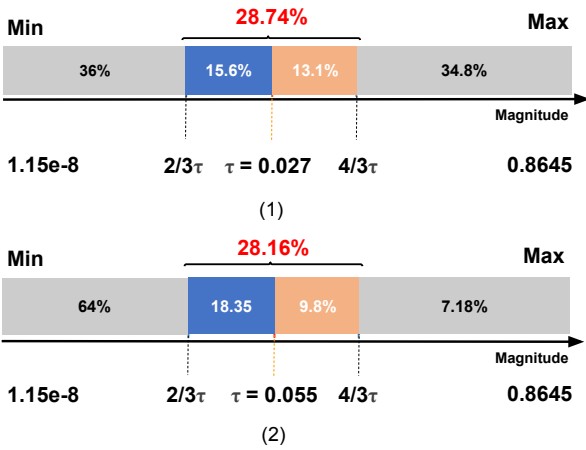

Figure 1: Weight Distribution in a Feedforward Layer of $BERT_{Base}$ at Various Sparsity Levels (0.52 and 0.83), Corresponding to Pruning Thresholds $\tau = 0.027$ and $\tau = 0.055$. Notably, around 29% of the weights lie within the range $[\frac{2}{3}\tau, \frac{4}{3}\tau]$. This observation puts into question the efficacy of magnitude-based pruning, as these weights, despite their proximity to the threshold, might play a crucial role in maintaining the model's accuracy. This suggests that directly eliminating weights with smaller magnitudes could potentially lead to a suboptimal pruning strategy.

2020), namely to remove the elements with the smallest magnitude. Here the magnitude refers to not only the weights but also the output sensitivity, gradients, or Hessian matrices of the training loss (Luo et al., 2017; Yu et al., 2018; He et al., 2019; Lis et al., 2019; Molchanov et al., 2019; Singh and Alistarh, 2020; Dong et al., 2017). While magnitude-based pruning can generate state-of-the-art results in a wide range of tasks, its pruning strategy is deterministic and solely depends on a single criterion, which lacks variety (we demonstrate this more thoroughly in the next paragraph). Furthermore, magnitude-based pruning is proven not optimal at high-level sparsity (Sanh et al., 2020). To further improve the pruning performance, Zhuang et al. (2020); Ge et al. (2011); Savarese et al. (2020); Verdenius et al. (2020);

Azarian et al. (2020) try to enlarge the search space of sparse architecture with regularization-based methods, which are non-deterministic. They design fine-designed $L_0$ or $L_1$ penalty terms added to the loss function. In this way, the model proactively shrinks some of the weights until they don't contribute to the final loss. Regularization-based methods can achieve noticeably better results than magnitude-based methods, especially at high-level sparsity (Sanh et al., 2020). However, this line of work often suffers from a non-convex landscape and is challenging to optimize with extra hyperparameters. In parallel, Su et al. (2020); Liu et al. (2022) adopt a more aggressive strategy to prune elements in a completely random way. Their methods demonstrate the competitive effect in small datasets (such as CIFAR-10, CIFAR-100 (Krizhevsky et al., 2009)) but fail in large datasets (such as ImageNet (Deng et al., 2009)). Different from these works, this paper introduces a mildly random pruning method that brings a controllable degree of randomness into the pruning mask generation procedure.

We demonstrate the weakness of magnitude-based pruning in Figure 1. It presents the weight distribution of a feedforward layer of BERT (Devlin et al., 2019). Define $\tau$ as the pruning boundary. We consider two scenarios: $\tau = 0.027$ and $\tau = 0.055$, leading to sparsity levels of 0.52 and 0.83, respectively. As shown in Figure 1, a large portion of weights ($\approx 29\%$) falls into the range of $[\frac{2}{3}\tau, \frac{4}{3}\tau]$, which cannot be overlooked because that it is unclear if the pruned weights close to the threshold contribute less than the kept weights in the final accuracy. The weights with smaller magnitudes can still be crucial, especially when dealing with edge cases or infrequent situations. Proximity between weights can intensify decision-making challenges. This is why the direct removal of weights with smaller magnitudes is sub-optimal, as also demonstrated in Gomez et al. (2019). Based on the above observations, we investigate the following questions in this paper:

**Question 1.** *Which is better for pruning? a deterministic way or a randomized way*?

Previous literature has not reached a consistent conclusion. While Su et al. (2020) and Liu et al. (2022) have provided evidence that random pruning can yield competitive or better results compared to deterministic methods, this finding does not consistently hold true for larger datasets. Moreover,

these results have not been universally extended to language models. We conjecture that their methods introduce unbridled randomness but do not provide any effective negative feedback. Moreover, exploring the extent of introduced randomness in a principled way has also not been explored in the previous literature. In this paper, we study and extend the above question systematically.

**Question 2.** *Can we design a consistently effective randomized pruning method*?

This paper answers the above question with the following contribution. **First**, we propose a randomized pruning mask generation strategy that can introduce controllable randomness in a principled way. **Second**, we design Mask Candidate Selection Strategy (MCSS) to choose the optimal mask from the pool of mask candidates, ensuring the introduced randomness always guides pruning in a beneficial direction. **Third**, to further enhance efficiency, we introduce Early Mask Evaluation Pipeline (EMEP) to mitigate the overhead associated with training under multiple pruning masks. **Last**, we offer an empirical guidance for randomized pruning on $\text{Bert}_{base}$ and $\text{Bert}_{large}$. Our results show a consistent accuracy boost ($\textbf{0.1\%}\sim\textbf{2.6\%}$) on the GLUE benchmark, outperforming other state-of-the-art pruning techniques at a 16x compression rate. Notably, our approach showcases even more significant enhancements ($\textbf{2\%}\sim\textbf{4\%}$) at extreme sparsity levels like the 100x compression rate.

## 2  Preliminaries

### 2.1  Pruning

**Iterative Magnitude Pruning**  Iterative Magnitude Pruning (IMP) is the most well-known strategy because it yields state-of-art results than others (Frankle and Carbin, 2019; Frankle et al., 2020), such as Single-shot Network Pruning (SNIP) (Lee et al., 2018). Specifically, we divide the pruning process into multiple stages by gradually increasing the sparsity. In each stage, pruning is to find and eliminate redundant parameters or neurons at that time. The most intuitive approach is to assign an importance score to each element and keep only the top-k elements. The score used to rank elements can be the absolute value of weights, output sensitivity, gradients, or other fine-designed metrics (Hagiwara, 1993; Gale et al., 2019; Thimm and Fiesler; Han et al., 2015; Zhu and Gupta, 2017; Cuadros et al., 2020; Luo et al., 2017). In this work,

different from the traditional deterministic way, we extend the IMP in a random way.

## 2.2 Knowledge Distillation

Knowledge Distillation (KD) (Hinton et al., 2015) is another compressing technique trying to transfer the knowledge from a well-trained large model T to a small model S. Many previous works have proved that pruning with KD can significantly reduce accuracy loss for Transformer-based models (Xu et al., 2021; Xia et al., 2022). Our experiments evaluate the pruning methods based on BERT (Devlin et al., 2019), and we apply the KD method to both the baseline and our strategy. Specifically, we distill the knowledge from the hidden state of each transformer block and the attention score of each self-attention layer. Figure 6 demonstrates the distillation strategy in our settings.

## 2.3 Multinomial Distribution

In probability theory, a multinomial distribution describes the probability distribution of $n$ $(n > 2)$ sampling trials from elements with $k$ $(k > 2)$ categories, which is a generalization of the binomial distribution (Ross, 2010). In our setting, the number of categories equals the number of elements, and the target sparsity and the total number of elements determine the number of trials. Note that the sampling process in this paper is done without replacement. This kind of sampling is also referred to as sampling from a multivariate hypergeometric distribution (Berkopec, 2007).

## 3 Methodology

In this section, we first rethink the traditional deterministic pruning method and introduce our basic idea and method. Following that, we elaborate on the details of our randomized pruning mask generation and selection strategy. The architecture of our strategy is depicted in Figure 2, and the detailed procedure is outlined step-by-step in Algorithm 1.

## 3.1 Rethink Iterative Magnitude Pruning

Traditional IMP divides pruning into multiple stages and generates a deterministic pruning mask by retaining the top-k elements at each stage. This process is based on the assumption that the top-k elements contribute more than the removed part. However, given the complex topology of model architecture and observations from Figure 1, it is difficult to draw such a conclusion. In this paper,

we aim to introduce a certain degree of randomness into the process of pruning mask generation, thereby expanding the search space for locally optimal pruning masks at each stage. Specifically, we propose a strategy for generating and selecting randomized pruning masks at each pruning stage.

## 3.2 Randomized Pruning Mask Generation

### 3.2.1 Mask Sampling

Different from the deterministic mask generation, we seek to infuse controllable randomness into this process. In essence, our approach is to sample the retained elements from a multinomial distribution without replacement. Specifically, the first step is to derive a probability distribution by normalizing the magnitude of the elements. Within our framework, magnitude is defined as the absolute value of the weight. Subsequently, we sample $k$ indices from this distribution, where $k$ represents the number of elements retained. Finally, we generate a binary mask, with locations corresponding to these indices set to one, effectively outlining the sparse architecture of the model. The utilization of this approach offers a refreshing departure from the deterministic way and creates a larger optimization space for model pruning.

### 3.2.2 Controllable Randomness

We have proposed a random method to generate pruning masks. However, for current models with several million parameters per layer, a single iteration of sampling leads to considerable randomness due to the minute probability post-normalization. To quantify this randomness, we propose $ir$ (introduced randomness) in Equation 1:

$$ir = (C * sparsity - C_s)/C_s \qquad (1)$$

Here, $C$ and $C_s$ represent the total count of weights and the count of weights pruned by both deterministic and random approaches, respectively. A small value of $ir$ indicates that the sampled mask resembles the deterministic one. Conversely, a larger value suggests a noticeable departure from the deterministic method.

We assess the introduced randomness with $ir$ and simultaneously strive to regulate its quantity. Drawing inspiration from the concept of model soup (Wortsman et al., 2022), we manage the randomness by sampling $M$ masks and adding them element-wise to craft an ensemble mask. This mask has its top-k values set to 1, with the remainder set to 0, thus yielding a mask with controllable

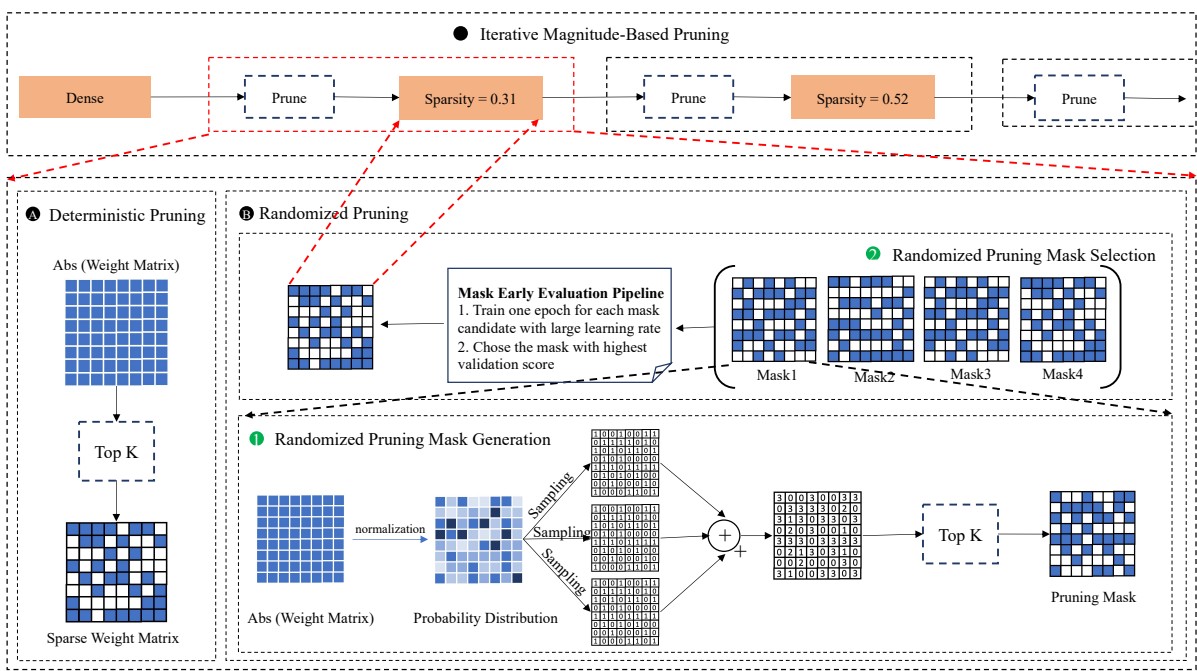

Figure 2: Main Architecture of Our Strategy. We replace **Ⓐ** the deterministic mask generation way in IMP with **Ⓑ** our randomized method. Specifically, **❶** we first introduce a degree of randomness into the process of mask generation in a principled way, **❷** then we employ a specific mask selection rule, paired with an efficient evaluation pipe, to distinguish the optimal mask from a pool of candidates.

randomness (k is the number of kept elements). Importantly, the degree of introduced randomness shows a positive correlation with the value of $M$.

### 3.2.3 Accelerated Mask Sampling

Controlling randomness solely by increasing the number of sampled masks can be time-intensive. To address this, we suggest deriving the sampling probability distribution from the $w^T$, where $w$ is the weight of the corresponding layer. In this scenario, the power value $T$ in the exponential term is used to control the variance of the sampling probability. As $T$ increases, the sampling probabilities for larger and smaller magnitudes diverge more, allowing Mask Sampling to curtail the introduced randomness swiftly. Moreover, aligning with our motivation to introduce randomness in mask generation, we only sample weights whose magnitudes are close to the pruning boundary $\tau$. We introduce more details in Appendix.

### 3.3 Randomized Pruning Mask Selection

### 3.3.1 Mask Candidate Selection Strategy

Our sampling approach expands the search space for locally optimal masks compared to the deterministic way. However, this inadvertently introduces undesired noise, leading to poor model accuracy because we introduce randomness without providing any effective negative feedback to the model optimization. To address this, we propose Mask Candidate Selection Strategy (MCSS) to ensure the introduced randomness always guides the model optimization in a beneficial direction. Specifically, at each pruning stage, we generate $N$ candidate masks and select the best one for the next pruning stage. To ensure robustness in our approach, we adopt a deterministic mask as one of our mask candidates. By doing so, we are not solely relying on random or heuristic methods but also have a reliable fallback.

### 3.3.2 Early Mask Evaluation Pipeline

To accelerate the mask selection, we design Early Mask Evaluation Pipeline (EMEP) to reduce computational costs. Specifically, we only fine-tune the model one epoch with a large learning rate for each candidate mask. The candidate that achieves a superior early-stop evaluation metric on the validation dataset is then deemed the winner. We crafted this strategy based on findings by Li et al. (2019) and You et al. (2019), which suggest that using a high learning rate during the earlier optimization iterations can yield a good approximation of the sparse network structure. Once the winner has

**Algorithm 1:** Randomized Pruning Mask Generation and Selection

```
Input: w ;                  /* one layer weight */
Result: w, M;                          /* mask */
s ← [s₁, s₂, ...] ;         /* pruning schedule */
sr ← 0.00005 ;                 /* sampling ratio */
n ← 8 ;                       /* # of candidates */
train w ;                        /* dense model */
foreach sₜ ⊂ s do
    for i ← 0 to n do
        p ← |w|/∑|w| ;
        p ← p⁵ ;                /* sampling prob */

        x ← sₜ × num_w ;  /* x is # of zeros
          in w_j after pruning */
        k ← num_w − x ;           /* k is # of
          non-zeros in w_j after pruning */

        M_i ← zeros_like w;
        m ← zeros_like w;

        y ← int(sr × x);
        for _ ← 0 to y do
            m ← 0
            pos ← sampling k positions from p;
            m[pos] ← 1;
            M_i ← M_i + m
        end
        M_i[top_k] ← 1; otherwise 0

        w = w × M_i;
        finetune one epoch with large lr;
        metric_i ← evaluate validation dataset;
    end
    select M with best metric;
    rewind w and lr;

    w = w × M;
    finetune w ;                    /* sparse model */
end
```

been chosen, we revert the weights and learning rate to their state before the last pruning step. Subsequently, the winning candidate mask is employed for continuous regular training.

# 4 Experiments

We evaluate the effectiveness of our pruning strategy in a wide range of natural language understanding tasks. Following previous work, we use **BERT** as our backbone and then apply different pruning methods to compare their performance.

## 4.1 Baselines

$BERT_{base}$ and $BERT_{large}$ are first chosen as our baseline models. Based on them, we apply **IMP** to generate 16x sparse models. These sparse models are used as our main baselines. In addition, we compare our strategy with previous works that have reported results on the same datasets, which including: **BERT-PKD** (Sun et al., 2019), **Stru-Pruning**$_{Roberta}$ (Wang et al., 2020), **SNIP** (Lin et al., 2020), **EBERT** (Liu et al., 2021), **BERT-of-Theseus** (Xu et al., 2020), **EfficientBERT** (Dong et al., 2021), **Sparse-BERT** (Xu et al., 2021), **RPP** (Guo et al., 2019), **Pretrained Ticket** (Chen et al., 2020), **Lottery Ticket** (Prasanna et al., 2020), **Prune at Pre-training** (Gordon et al., 2020), **Movement Pruning** (Sanh et al., 2020), **DistillBert**$_6$ (Sanh et al., 2019), and **TinyBert**$_6$ (Jiao et al., 2020).

## 4.2 Datasets and Data Augmentation

Following previous works, we select eight tasks from the **GLUE** dataset (excluding **WNLI**) to evaluate the effectiveness of our pruning strategy (Wang et al., 2018). We also follow the data augmentation method from **TinyBert** (Jiao et al., 2020). More details can be found in Appendix.

## 4.3 Setup

We follow the strategy from SparseBert (Xu et al., 2021) to do pruning and knowledge distillation simultaneously at downstream tasks. We also imitate the setting from (Frankle and Carbin, 2019) to adopt a simple pruning schedule in our experiments. Specifically, we only use 4-9 pruning stages to increase the sparsity gradually (such as 0.54, 0.83. 0.91, 0.9375). Furthermore, after choosing a decision mask at each pruning stage, the number of epochs in the finetuning phase is no longer limited until the model converges. We apply exactly the same setting for the IMP baseline and our approach. For more details about hyperparameters, please refer to Appendix.

## 4.4 Main Results and Analysis

We report the results on *dev* set of 8 datasets from GLUE and summarize them in Table 1-2. We also compare our results with distillation-based methods and describe the results in Appendix.

From the above results, we can easily observe that our strategy consistently generates better performances than the main baseline IMP in a totally same setting. Moreover, the result of our method is also optimal in similar settings compared to other pruning techniques. These findings demonstrate the superiority of our random way in the mask generation process over the deterministic approach and confirm that our mask selection rule can effectively navigate the optimization after introducing randomness. In other words, our methods successfully increase the probability of finding better pruning

| Methods | #params | MNLI-m Acc | QNLI Acc | QQP F1/Acc | MRPC F1 | SST-2 Acc | COLA Mrr | RTE Acc | STS-B Spear | |
|---|---|---|---|---|---|---|---|---|---|---|
| BERT$_{Base}$ | 110M | 84.5 | 91.4 | 89.59/91.0 | 90.1 | 92.5 | 56.3 | 69.3 | 89.0 | |
| *left #params $\geq$ 50%* | | | | | | | | | | |
| BERT-PKD | 50% | 81.3 | 88.4 | -/88.4 | 85.7 | 91.3 | 45.5 | 66.5 | 86.2 | - |
| Stru Pruning | 73% | - | 89.05 | - | 88.61 | 92.09 | - | - | 88.18 | |
| SNIP | 50% | 82.4 | 89.5 | - | 88.1 | 91.8% | - | - | - | |
| EBERT | 60% | 83.1 | 90.2 | 87.5/90.8 | - | 92.2 | - | - | - | |
| BERT-of-Theseus | 50% | 82.3 | 89.5 | -/89.6 | 89.0 | 91.5 | 51.1 | 68.2 | 88.7 | |
| Pretrained Ticket | 50%-90% | 82.6 | 88.9 | -/90.0 | 84.9 | 91.9 | 53.8 | 66.0 | 88.2 | |
| Lottery Ticket | 38%-51% | 84.0 | 91.0 | -/91.0 | 84.0 | 92.0 | 54.0 | 61.0 | 88.0 | |
| IMP | 50% | 84.6 | 91.3 | 88.0/91.0 | 90.8 | 92.8 | 53.1 | 72.0 | 89.4 | |
| **Ours** | 50% | **84.7** | **91.5** | **88.1/91.1** | **91.5** | **93.0** | **54.3** | **72.3** | **89.5** | |
| *left #params $\leq$ 10%* | | | | | | | | | | |
| RPP | 10% | 78 | 87 | 88.0/- | 80.0 | 89 | - | - | - | |
| Movement Pruning | 10% | 80.7 | - | 87.1/90.5 | - | - | - | - | | |
| EfficientBERT | 9% | 81.7 | 89.3 | 86.7/- | 90.1 | 90.1 | 39.1 | 63.2 | 79.9 | |
| SparseBERT | 5% | - | 90.6 | - | 88.5 | - | 52.1 | 69.1 | - | |
| IMP | 6% | 83.3 | 90.5 | 87.6/90.8 | 90.2 | 92.2 | 53.1 | 66.7 | 87.0 | |
| **Ours** | 6% | **83.4** | **90.9** | **87.9/90.9** | **91.5** | **92.7** | **53.4** | **69.3** | **87.5** | |

Table 1: Main Comparison Results between Our Strategy and Other Pruning Baselines with Bert$_{Base}$ on *dev* Sets of 8 Datasets from GLUE Benchmark. Note that the pruning results of IMP and **Ours** are achieved by ourselves in a totally same setting, while others are from the corresponding literature.

masks by introducing randomness in a principled way.

We also notice that the potential improvement of performance may be limited by the magnitude used to derive sampling probability. In our setting, we use the absolute value of weights to decide the importance of each neuron connection. Thus our pruning results can not surpass the theoretical optimal result (upper bound) when pruning with the absolute value of weights. This reminds us that our method can be easily transplanted to other magnitude-based pruning methods, such as the gradients-based methods, and may produce the same effect, helping us find better pruning masks.

Furthermore, we realize that the effect of our strategy in different datasets is not uniform. We obtain a more noticeable improvement in accuracy on small datasets. We argue that small datasets have more local minimums of loss surface, and therefore our strategy can more easily help find better pruning masks.

## 4.5 Ablation Study

We try different ablation settings to figure out the functionality of each part of our strategy and analyze why they are effective for model pruning.

### 4.5.1 Impact of Randomness and Schedule

Prior studies have demonstrated that there is no loss in accuracy at lower levels of sparsity, particularly when sparsity is less than 50%. This suggests that the model maintains robustness with the architectures identified in the early stages of pruning. We conjecture there is a high level of redundancy in the weights at the early pruning stage. As such, introducing more randomness could potentially expand the search space for sparse architecture in the early stages without hurt to the accuracy. However, previous works also argue the concept of early model adaptions and emphasize the importance of early searched architecture to the final pruning targets. Thus, introducing too much randomness at the beginning stage may not be a good idea.

In the last pruning stage, the closely-matched magnitudes of the remaining weights significantly impact accuracy. Careful selection is required for the elements to be pruned. Too much randomness could corrupt the model, hindering recovery from the last pruning step, while too little might restrict our search for potentially optimal masks. Hence, deciding the best schedule of randomness at each pruning stage is crucial.

| Methods | #params | MNLI-m Acc | QNLI Acc | QQP F1 | MRPC F1 | SST-2 Acc | COLA Mrr | RTE Acc | STS-B Spear | |
|---|---|---|---|---|---|---|---|---|---|---|
| BERT$_{Large}$ | 330M | 86.6 | 92.3 | 91.3 | 89.1 | 93.2 | 60.6 | 74.8 | 90.0 | |
| IMP | 20% | 85.2 | 91.6 | 90.8 | 90.9 | 92.8 | 59.0 | 73.2 | 89.1 | |
| **Ours** | 20% | **86.2** | **91.8** | **91.1** | **91.9** | **93.7** | **60.9** | **75.5** | **89.9** | |

Table 2: Comparison between Our Strategy and IMP with BERT$_{Large}$ on GLUE *dev* Sets.

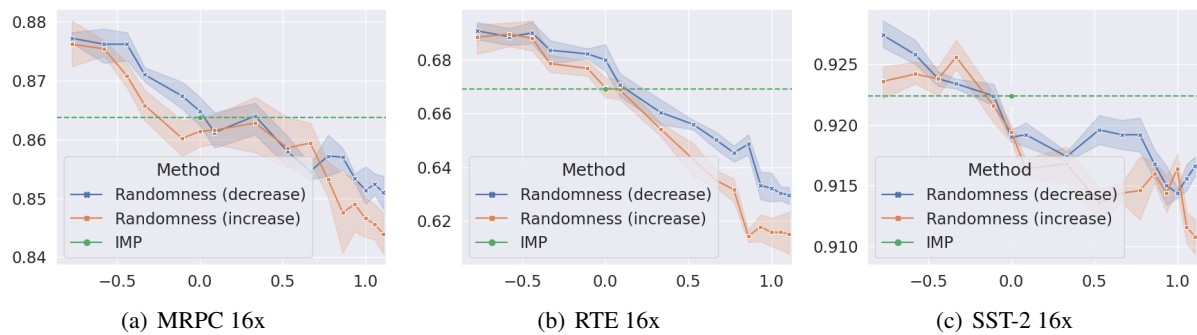

(a) MRPC 16x      (b) RTE 16x      (c) SST-2 16x

Figure 3: Comparing the Impact of Randomness in Two Different Schedules with a Deterministic Approach (IMP), which features zero randomness. The horizontal axis presents the logarithmic outputs of $ir$, with larger $ir$ indicating a greater amount of total introduced randomness. The vertical axis signifies the model's accuracy.

To investigate the impact of randomness, we introduce a hyper-parameter $sr$ that controls the number of sampled masks $M$, and thereby controls the total introduced randomness. We also propose two simple randomness schedules at varying pruning stages: (1) **Decrease**, where we reduce the introduced randomness by increasing the number of sampled masks as the count of pruned weights increases ($M = sr * C_{pruned}$), and (2) **Increase**, where we enhance the introduced randomness by decreasing the number of sampled masks as the count of pruned weights increases ($M = sr * (C - C_{pruned})$).

We conduct experiments comparing these two schedules against our primary baseline (IMP) under different $sr$ values. The results are displayed in Figure 3, leading us to the following observations: 1) Excessive randomness results in our strategy performing even worse than the deterministic method (the region blue and orange lines below the green line). In this region, the *Decrease* strategy outperforms the *Increase* strategy. 2) Existing a threshold below which both our two randomness schedules outperform IMP, highlighting the superiority of our random approach over the deterministic way. 3) Existing another threshold above which the performances of the two randomness schedules become virtually identical. 4) The *Decrease* strategy consistently equals or outperforms the *Increase* strategy.

This proves that the model accuracy is not sensitive to randomness in the early pruning stage and is gradually becoming sensitive as it approaches target sparsity.

### 4.5.2 Impact of MCSS

We assessed the role of MCSS by removing it from our strategy and comparing the results with our primary findings. The results are summarized in Figure 4. We make the following observation: 1) In the setting with MCSS, there is a certain threshold of randomness below which our strategy significantly outperforms the deterministic way. 2) In contrast, In the setting without MCSS, the model's performance against the deterministic approach is inconsistent and lacks a clear trend or pattern. This precisely demonstrates that MCSS can ensure the introduced randomness consistently guides the model optimization toward a beneficial direction. In other words, MCSS effectively enhances the lower accuracy boundary in our experiments.

### 4.5.3 Impact of Sparsity

We have examined our strategy across various levels of sparsity, and the findings have been encapsulated in Figure 5(a). We observe that our random pruning strategy has consistently demonstrated superior performance compared to the baseline (IMP) across all levels of compression. The advantage is particularly pronounced at higher levels of sparsity,

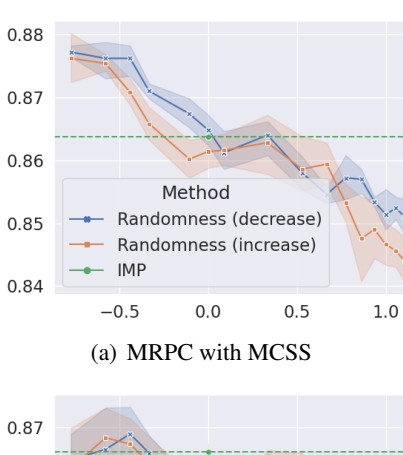

(a) MRPC with MCSS

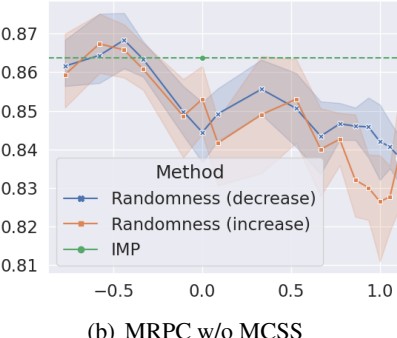

(b) MRPC w/o MCSS

Figure 4: Mask Sampling 4(a) v.s. Mask Sampling + MCSS 4(b). Note that the green line in 4(a) and 4(b) represents the same value of accuracy from IMP. The value on the horizontal axis represents the amount of introduced randomness. The value on the vertical axis indicates model accuracy.

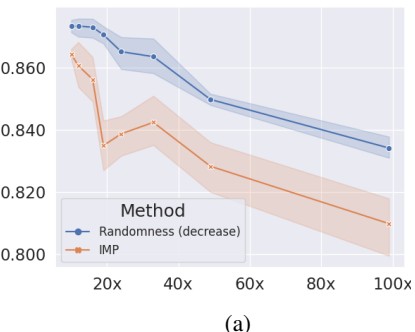

(a)

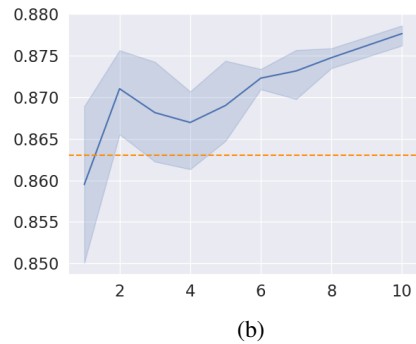

(b)

Figure 5: Impact of Sparsities 5(a) and Impact of #Mask Candidates in MCSS 5(b). The horizontal axes represent the sparsity level and the number of mask candidates for Figures 5(a) and 5(b) respectively, while the vertical axes in both figures denote model accuracy.

such as those equal to or greater than 16x sparsity.

### 4.5.4 Impact of Masks Candidates

To verify the relationship between the number of mask candidates in MCSS with the final performance, we design experiments to increase the number of candidate masks at each pruning stage from 2 to 10, and the results are depicted in Figure 5(b). We conclude a positive correlation between the quality of the searched mask and the number of candidate masks at each stage. As the number of mask candidates close to 10, the performance gain is gradually missing. Additionally, the variance of the performance gain is also gradually minimized, which proves that the MCSS can effectively navigate the optimization of pruning in a beneficial direction.

### 4.6 Discussion

#### 4.6.1 Efficiency Analysis

We analyze the efficiency of our method through the training and inference phases.

**Training Phase:** In the training phase, we compare the required computation of our method with the traditional IMP method. It's easy to find that

additional computation mainly from the randomized mask generation and selection. For the generation process, it's crucial to emphasize that the creation of each mask is independent of the others. This independence allows for parallel processing, meaning that the time consumption doesn't increase linearly as the increase of mask candidates. On the other hand, we measured the GFLOPS required to generate a single mask and compared it with the GFLOPS needed for one forward pass of $BERT_{base}$. It's roughly 1 percent to the later operation. However, due to implementation challenges, we couldn't concurrently sample $k$ positions multiple times from the weight matrix, leading to an overall increase in processing time for single randomized mask generation. For the selection process, we only require one epoch to identify the optimal mask, where the overhead is minimal compared with the entire pruning process.

**Inference Phase:** In real-world applications, although there might be overheads during training, the benefits reaped during inference make it worthwhile. Our method stands out with a 16x compression rate and can sustain performance even at

higher sparsity levels, achieving up to a 100x compression rate. This ensures that our pruned neural networks, once deployed, bring about significant improvements in performance and efficiency.

### 4.6.2 Extending to Billion Parameters

In the current age of large language models, achieving effective pruning is a formidable challenge, particularly when striving to preserve high sparsity without sacrificing performance. While initiatives like SparseGPT have ventured into pruning for these colossal models, they have only managed a 2x compression rate (Frantar and Alistarh, 2023). The computational complexity of our method is primarily determined by the number of parameters involved. Consequently, our random pruning technique is not yet adaptable to models with billions of parameters. Nevertheless, we are diligently working on refining methods that incorporate controllable randomness more efficiently.

## 5 Related Work

A number of researchers have explored pruning in BERT. Prasanna et al. (2020) prunes the model with Michel et al. (2019)'s first-order importance metric and proves that unstructured magnitude-based pruning always produces sparser and higher-accuracy models than structured pruning. Gordon et al. (2020) and Chen et al. (2020) both argue that pruning in the pre-trained stage is better than fine-tuning stage because there is no need to prune for each downstream task. They also conclude that knowledge in sparse training can be transferred as well as dense models. In contrast, Xu et al. (2021) find that pruning at the pre-trained stage has huge computational problems while pruning at the finetuning stage can save computational efforts and keep accuracy simultaneously. These pruning methods are based on the magnitude and prune weights in a deterministic way. In parallel, a linear of works defeats magnitude-based methods at high-level sparsity by applying a non-deterministic way: regularization-based pruning. Specifically, a fine-designed $L_0$ or $L_1$ penalty terms are added to the loss function. Then the model proactively shrinks some of the weights until they do not contribute to the final loss. The regularization-based method can generally achieve significantly better results than the magnitude-based methods, especially at high-level sparsity (Sanh et al., 2020). However, penalty terms can introduce additional local minima to the

loss function and are difficult to navigate optimization. On the other hand, there is a lack of research on random pruning applied to Transformer-based models (such as BERT) in previous studies. Therefore, our paper complements the gap in this area.

## 6 Conclusion

This paper presents a non-deterministic model pruning method, introducing controllable randomness by generating binary masks in a specific random fashion. Coupled with our specific mask candidate selection rule, our method exhibits significant effectiveness in enhancing model accuracy in pruning, particularly at high levels of sparsity.

## Limitations

Previous random pruning techniques do not scale to large datasets probably because the search space is too large to find a winning pruning mask. There's a considerable likelihood that our method isn't flawless either. For example, a more fine-designed randomness schedule could potentially yield more substantial benefits. In addition, our method might not be appropriate for models with billions of parameters due to the cost of training under multiple pruning masks. A potential solution could be to execute this process in parallel since pruning masks are independent of each other.

## Acknowledgements

The authors wish to thank the anonymous reviewers for their helpful comments.

## Ethics Statement

This research is conducted in compliance with the ACL Ethics Policy. In the pursuit of advancing the efficiency of model pruning strategies, our methods raise several ethical considerations. First, our research proposes a new model pruning strategy that significantly enhances the efficiency of neural networks, which could potentially contribute to the democratization of artificial intelligence by making it accessible to systems with lower computational resources.

However, despite these potential benefits, the increased efficiency and performance of neural networks might lead to misuse if not properly regulated. For instance, such improvements could potentially contribute to the spread of deepfake technology, an area of AI that has been used to

spread disinformation and cause harm. It is important to note that our research is conducted with the intent of improving efficiency and accessibility in AI, and we explicitly denounce any misuse of our work for harmful purposes. We encourage further discussions on the implementation of safeguards to prevent misuse and to guide the ethical use of such technology.

In addition, we must consider the implications of our method on fairness and bias. While our work is focused on model efficiency, the datasets used in the process may contain inherent biases, which can result in the propagation of these biases in pruned models. We believe it is essential to ensure that data used in our processes is as unbiased and representative as possible, and to strive for transparency in this regard.

Lastly, we acknowledge that the reduction of model size may result in the loss of interpretability, as smaller models often have more complex and less interpretable decision-making processes. We urge researchers and practitioners to maintain a balance between model efficiency and transparency to ensure fair and explainable AI practices.

We hope that our work will stimulate further discussions around these considerations, and that it will encourage future research to continuously consider the ethical implications of their work.

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

# A  Appendices-A

## A.1  Model Compression

Transformer-based models are proven effective both in natural language processing and computer vision tasks (Dosovitskiy et al., 2020; Raffel et al., 2020). However, this series of models is restricted by its massive memory storage and computational cost at inference: for example, BERT (Devlin et al., 2019) and GPT-3 (Brown et al., 2020) are hardly possible to deploy in practical scenarios, with no mention for edge devices. To alleviate this problem, many approaches have been invented to compress large language models, such as Knowledge Distillation, Parameter Sharing (Jiao et al., 2020; Sachan and Neubig, 2018), Quantization, and Model Pruning. Moreover, previous works also show that training a large but sparse model leads to better results than a small but dense model (Gomez et al., 2019). Therefore, pruning techniques have gained more attention than other compression techniques.

## A.2  Datasets and Data Augmentation

We select eight tasks from the GLUE dataset to evaluate the effectiveness of our pruning strategy. They are: CoLA (Warstadt et al., 2019), SST-2 (Socher et al., 2013), 3 sentence similarity tasks: MRPC (Dolan and Brockett, 2005), STS-B (Cer et al., 2017), QQP (Chen et al., 2018), and 3 natural language inference tasks: MNLI (Williams et al., 2018), QNLI (Rajpurkar et al., 2016), and RTE (Bentivogli et al., 2009). The metrics for these tasks can be found in the GLUE paper (Wang et al., 2018).

This paper also employs the data augmentation method proposed in TinyBert (Jiao et al., 2020). We have adopted a two-pronged approach for word-level replacement data augmentation by combining a pre-trained language model, BERT, with GloVe word embeddings (Pennington et al., 2014). For single-piece words, the BERT model is used to predict word replacements (Wu et al., 2019), while for multi-piece words, we rely on the GloVe word embeddings to identify the most similar words as potential replacements. The process of word replacement is guided by specific hyperparameters. These parameters control the proportion of words replaced within a sentence and the overall volume of the augmented dataset. In our methodology, we maintain the same hyperparameters as utilized in TinyBert (Jiao et al., 2020). For further details regarding these hyperparameters, we direct readers to the original TinyBert study (Jiao et al., 2020).

However, unlike TinyBert (Jiao et al., 2020), our application of data augmentation is limited to smaller datasets, namely MRPC, RTE, SST2, and CoLA. Conversely, the larger datasets MNLI, QNLI, and QQP are sizable enough for our experiments without the need for augmentation. Additionally, we enhance the STS-B dataset by merging it with the MNLI-m dataset.

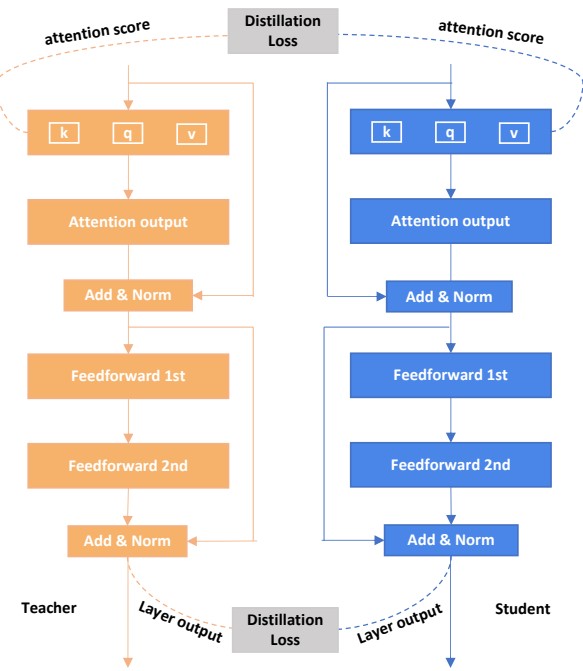

Figure 6: Details of KD in our experiments. In the pruning process, the sparse student model (S) continuously learns the knowledge from the dense teacher model (T).

## A.3  Compare with Distillation-based Models

In addition to comparisons with other pruning-based methods, our strategy also outperforms prevalent distillation-based methods, such as DistillBert$_6$ (Sanh et al., 2019) and TinyBert$_6$ (Jiao et al., 2020). The details of these comparisons are presented in Table 3.

# B  Appendices-B

## B.1  Acceleration of Mask Sampling

Our process involves sampling M pruning masks and summing them together to manage the introduced randomness. However, this method proved to be time-consuming. Given our original intent to incorporate randomness into the mask-generation process, our goal was to increase the retention probability of weights near the pruning boundary during

| models | #params | MNLI-m Acc | QNLI Acc | QQP F1/Acc | MRPC F1 | SST-2 Acc | COLA Mrr | RTE Acc | STS-B Spear |
|---|---|---|---|---|---|---|---|---|---|
| BERT$_{BASE}$ | 110M | 84.5 | 91.4 | 89.59/91.0 | 90.1 | 92.5 | 56.3 | 69.3 | 89.0 |
| DistillBert$_6$ | 50% | 82.2 | 89.2 | -/88.5 | 87.5 | 92.7 | 51.3 | 59.9 | 86.9 |
| TinyBert$_6$ | 50% | 84.5 | 91.1 | 88.0/91.1 | 90.6 | **93.0** | 54.0 | **73.4** | **89.6** |
| **ours** | 50% | **84.7** | **91.5** | **88.1/91.1** | **91.5** | **93.0** | **54.3** | 72.3 | 89.5 |

Table 3: Comparison between Our Strategy and Distillation-based Methods with BERT$_{Base}$ on GLUE *dev* Sets.

the pruning process. Consequently, we limit our sampling to target weights with magnitudes within the range of $[t_{top_{(r \cdot k)}}, m]$, where $t$ is the pruning boundary, $k$ is the number of weights retained, $m$ is the maximum magnitude, $r$ is a hyperparameter used to control the sampling range. Specifically, we assigned a zero sampling probability to weights with magnitudes less than $t_{top_{(r \cdot k)}}$. This method provided a more efficient and effective strategy for incorporating randomness into our pruning process.

### B.1.1 Randomness Schedule

Our method is supposed to sample $M$ masks and then add them together to control the introduced randomness. One left concern is how much randomness should be introduced for each pruning stage.

As introduced in Ablation 4.5.1, we propose two most straightforward sampling schedules to introduce randomness at different pruning stages: 1) *Decrease*, decreasing the introduced randomness by sampling more masks with the increase of pruned weights. 2) *Increase*, increasing the introduced randomness by sampling fewer masks with the increase of pruned weights. Specifically, a hyper-parameter $sr$ is introduced to control these two schedules. Equation 2 and Equation 3 demonstrate how to calculate the number of sampled masks for two strategies. Here, $M$ is the number of masks to be sampled, $sr$ is the sampling ratio, $C_{pruned}$ and $C_{kept}$ refers to the number of weights pruned and kept, respectively. In the iterative setting, with the sparsity reaching the target gradually, $C_{pruned}$ becomes larger, and $C_{kept}$ becomes smaller. Therefore, given the sampling ratio, *Increase* gradually increases the amount of imported randomness, while *Decrease* gradually decreases the amount of imported randomness.

$$C_{pruned} = C * sparsity$$
$$M = sr * C_{pruned} \tag{2}$$

$$C_{kept} = C - C * sparsity$$
$$M = sr * C_{kept} \tag{3}$$

### B.2 More Implementation Details

We provide additional details regarding the hyper-parameters utilized in our experiments. We follow the settings from Frankle and Carbin (2019), implementing a straightforward pruning schedule. Specifically, for the MRPC, RTE, and SST2 tasks, we employed [0.54, 0.83, 0.91, 0.9375] while for the MNLI-m, QNLI, QQP, STS-B, and CoLA tasks, the sequence [0.54, 0.83, 0.875, 0.9, 0.92, 0.9275, 0.93, 0.935, 0.9375] was adopted. Although a more complex pruning schedule, such as a cubic approach, might enhance the final performance, that is not our primary objective in this research.

We also use the hyperparameter $sr$ to manage the number of sampling masks $M$, choosing from several options (3e-6, 5e-6, 8e-6, 1e-5, 3e-5, 5e-5). Additionally, $T$, used in the exponential function to accelerate the reduction of introduced randomness, is set to 5. For each pruning stage, we selected 8-10 candidate masks. The learning rate remains constant during the pruning process yet linearly declines when the model achieves the targeted sparsity. The optimization process is managed by Adam, with a warm-up phase constituting 10% of the steps.

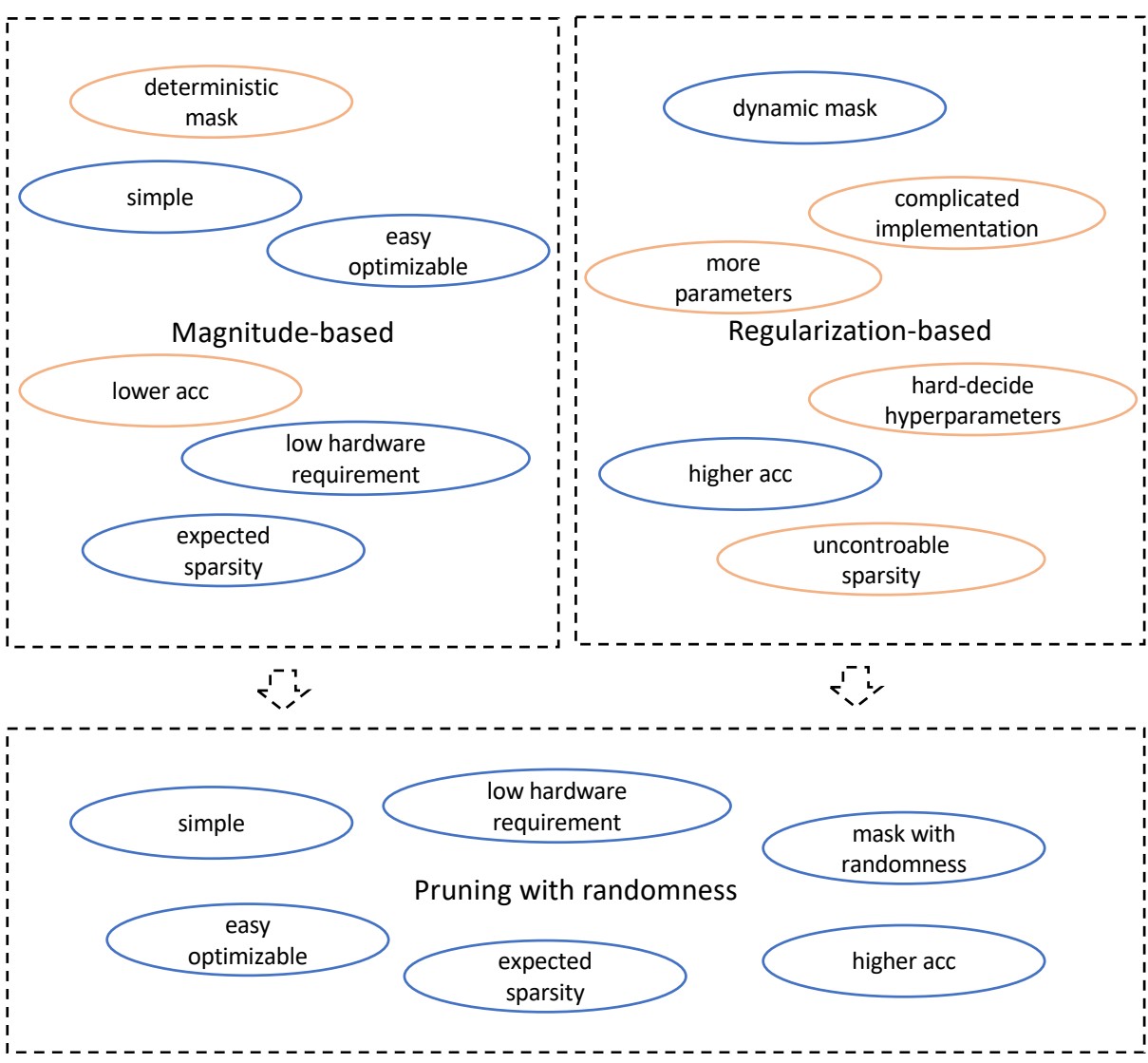

Figure 7: Pruning categories

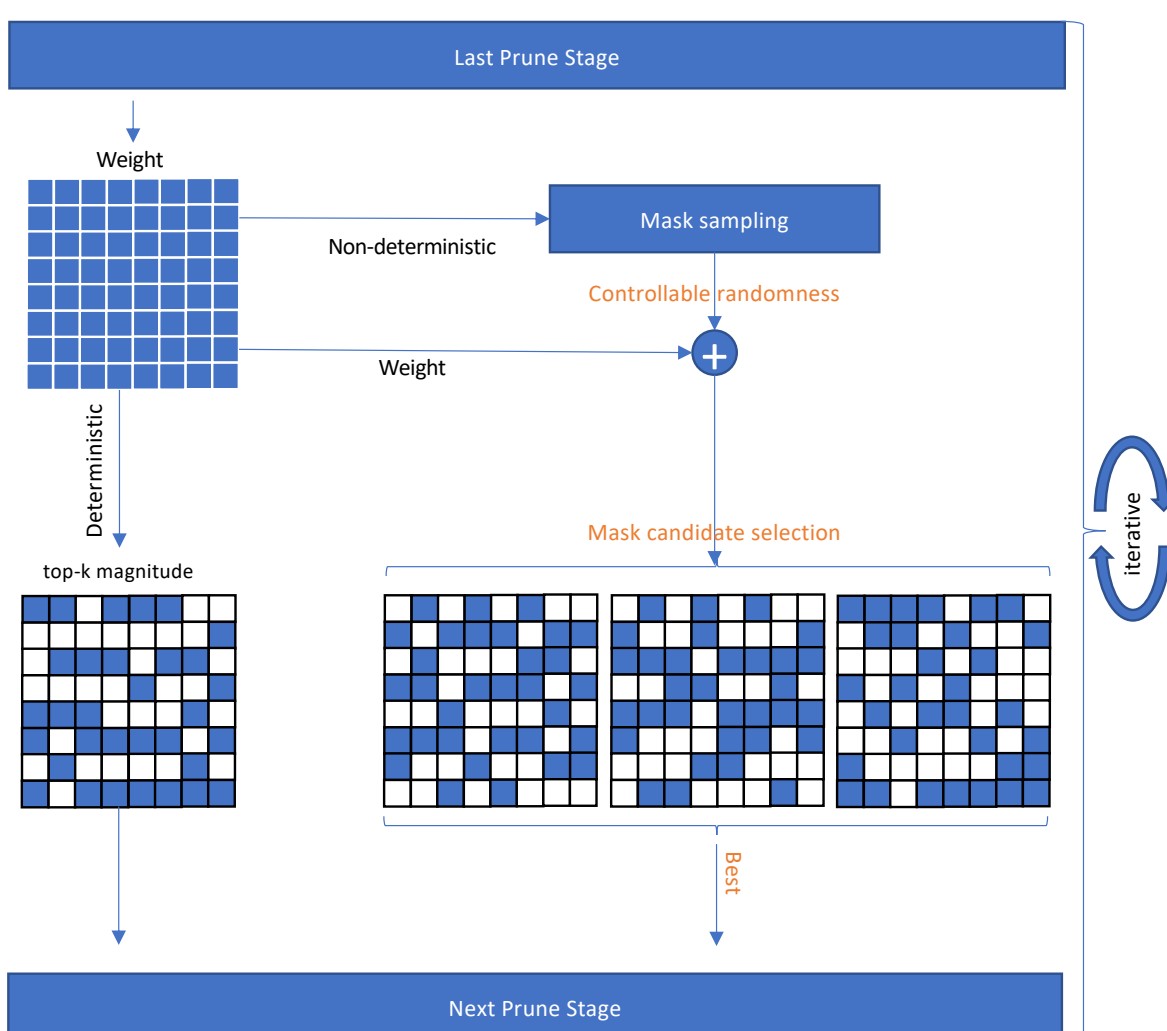

Figure 8: Deterministic Pruning **v.s.** Our Strategy