# OpenReview forum: "Breaking through Deterministic Barriers: Randomized Pruning Mask Generation and Selection"
_EMNLP/2023/Conference — EMNLP 2023 Findings_

### Official Review · Reviewer_gFqB · 2023-07-30

**Soundness:** 3

**Excitement:**

3: Ambivalent: It has merits (e.g., it reports state-of-the-art results, the idea is nice), but there are key weaknesses (e.g., it describes incremental work), and it can significantly benefit from another round of revision. However, I won't object to accepting it if my co-reviewers champion it.

**Paper Topic And Main Contributions:**

This paper proposes a model pruning strategy that first generates several pruning masks in a designed random way. Along with an effective mask selection rule and an early mask evaluation strategy, the approach achieves state-of-the-art performance across eight datasets from GLUE.

**Questions For The Authors:**

1. How to evaluate and select the best one for pruning in MCSS? What if all of them are not good enough? Moreover, as more candidates lead to more complexity, I wonder the training cost of the proposed method as it is not reported and compared with other methods in the paper.
2. How does ir work in your methods?
3. Do you consider appling the method on larger models in addition to BERT?


**Reasons To Accept:**

1. The method proposes a randomized pruning mask generation strategy that can introduce controllable randomness in a principled way.
2. The method achieves consistent improvement on benchmark compared with other state-of-art pruning methods.

**Reasons To Reject:**

1. The phenomenon of how randomness outperforms determinism is not explained and analyzed in detail.
2. No deeper analysis and comparison about the proposed method and other methods.

**Reproducibility:**

3: Could reproduce the results with some difficulty. The settings of parameters are underspecified or subjectively determined; the training/evaluation data are not widely available.

**Reviewer Confidence:**

4: Quite sure. I tried to check the important points carefully. It's unlikely, though conceivable, that I missed something that should affect my ratings.

---

> ### Author Rebuttal · Authors · 2023-08-25
>
> We sincerely thank the reviewer for their insightful comments and questions, which have offered us a clear perspective on areas for improvement. We have carefully gone through the feedback and summarized the major points raised. Below, we have listed the primary concerns and questions from the review to address them systematically in our rebuttal.
>
> Here is a summarized list of the questions or concerns extracted from the content:
>
> **Question 1**: The phenomenon of how randomness outperforms determinism is not explained and analyzed in detail
> **Question 2**: No deeper analysis and comparison with other methods.?
> **Question 3**: How to evaluate and select the best one for pruning in MCSS? What if all of them are not good enough?
> **Question 4**: How efficient is your method?
> **Question 5**: How does ir work in your methods?
> **Question 6**: Do you consider applying the method on larger models in addition to BERT?
>
> Please see our responses below to clarify the main concerns. Also, if you feel that your original concerns have been resolved, we would appreciate it if you would update your evaluation to reflect this. Thank you!
>
> ### Question 1: The phenomenon of how randomness outperforms determinism is not explained and analyzed in detail.
>
> We have previously discussed this topic in our paper, specifically in **`[Section 4.5.1: Impact of Randomness and Schedule]`**. In this section, we delve into the subtleties of incorporating randomness during the pruning stages. However, we appreciate the opportunity to elaborate further.
>
> - **`[Line 388-398]` Redundancy in Early Stages:** The fundamental understanding revolves around the high level of redundancy in the weights during the early pruning stages. Such redundancy suggests that there's significant flexibility in weight selection without adversely impacting accuracy. By introducing randomness, we potentially expand the architecture search space, offering a chance to discover more optimal pruned architectures.
>
> - **`[Line 399-407]` Delicate Balance in Late Stages:** On the other hand, the latter stages of pruning demand a more delicate approach due to the closely matched magnitudes of the remaining weights. Here, the balance becomes pivotal:
>   * Excessive randomness could jeopardize the model's integrity, making post-pruning recovery challenging.
>   * A meager introduction of randomness might restrict the exploration of potential optimal masks, possibly limiting the pruning efficiency.
>
> Our dedicated experiments, visualized in Figure 3, specifically delve into this balance between randomness and determinism. Key observations from these experiments include:
>
> 1. **`[Line 429-432]` Thresholds of Randomness:** There's a notable threshold beneath which our randomness-infused strategies eclipse the deterministic method. In parallel, there's another threshold, beyond which the performances of our two randomness schedules seem to converge.
> 2. **`[Line 435-440]` Sensitivity Towards Randomness:** The `\textit{Decrease}` strategy corroborates that the model's accuracy isn't particularly susceptible to randomness during the early pruning phase. However, as pruning continues, this resilience wanes, making the model progressively more sensitive to randomness.
>
> To wrap up, we've endeavored to offer insights and empirical findings to illustrate how calibrated randomness can surpass deterministic pruning in certain scenarios. Recognizing the reviewer's request for a more profound analysis, we're committed to furnishing a more meticulous examination in the ensuing version of our manuscript, ensuring lucidity.
>
> ### Question 2: No deeper analysis and comparison with other methods.
>
> Our primary focus in the study was to understand the effect of randomness relative to pruning, **`especially when benchmarked against our deterministic baseline, IMP`.** We aimed to examine, at a granular level, how randomness interacts with pruning at different stages of the model's life cycle, as elucidated in **`[Section 4.5.1 Impact of Randomness and Schedule]`**.
>
> - **Robustness at Early Stages:** Existing research indicates that models maintain their accuracy even with a high degree of sparsity, especially when it's below 50%. This reinforced our hypothesis about the redundancy in early-stage weights. Therefore, introducing randomness at this juncture potentially aids in widening the architecture search space, possibly leading to more efficient pruned architectures.
>
> - **Tuning Randomness:** The tail-end of pruning is particularly delicate due to the intricate balance required between weight magnitudes. Here, the judicious introduction of randomness becomes crucial:
>   * Too much randomness might jeopardize the model's stability.
>   * Insufficient randomness may curtail our quest for optimal masks.
>
> Our experiments, particularly depicted in Figure 3, accentuate this delicate interplay **`between randomness and deterministic pruning`**. Key takeaways from our findings include:
>
> 1. **Performance Dips with Excessive Randomness:** There exists a region where undue randomness makes our strategy lag behind the deterministic approach. Interestingly, in this very region, the **`Decrease`** strategy consistently surpasses the **`Increase`** one.
> 2. **Superiority of Random Approach:** We identified a threshold wherein our randomness-driven strategies consistently outperform IMP. This empirically underscores the potential benefits of integrating randomness into pruning.
> 3. **Convergence of Performance:** Beyond a certain threshold, the performance dynamics of our two randomness schedules become strikingly similar.
>
> With all due respect, while we deeply delved into the effect of randomness in pruning against our main baseline **`IMP`**, we admit that an exhaustive comparison with other methods wasn't the mainstay of this study.
>
> ### Question 3: How do you evaluate and select the best one for pruning in MCSS? What if all of them are not good enough?
>
> We evaluate masks using the validation dataset, testing their performance on unseen data. The best mask is selected based on this performance. As a precaution, we also incorporate a deterministic mask among our candidates, ensuring we don't solely depend on random or heuristic approaches. More details are provided below:
>
> 1. **Firstly**, as you pointed out, evaluating and selecting the best mask for pruning in MCSS is essential. To ensure robustness in our approach, we adopt a deterministic mask as one of our mask candidates. By doing so, we're not solely relying on random or heuristic methods but also have a reliable fallback.
>
> 2. **Secondly**, regarding the selection of the best mask, we conduct evaluations on the validation dataset (validation dataset). Doing this ensures that our model is tested on unseen data, providing an objective measure of the mask's performance. This process of selecting the best mask from multiple candidates allows us to mitigate the risk of all the masks performing sub-optimally.
>
> 3. **To directly address your concern**, "What if all of them are not good enough?", our deterministic mask acts as a safety net. Moreover, by evaluating multiple masks on the validation dataset, we largely ensure that our selection won't be compromised because all mask candidates perform poorly. This mechanism safeguards the quality of our selection during the pruning phase.
>
> ### Question 4: How efficient is your method?
> In response to your query about the efficiency of our method, we address this concern from two perspectives: the training and inference phases.
>
>   - **Training Phase**: Concerning the computational overhead of our method
>     * **Randomized Mask Generation:**
>          **(1)** At first glance, the computation in this phase might seem directly proportional to the number of randomized mask candidates we aim to generate.
>          **(2)** **`Utilizing Parallel Processing to Expedite Randomized Mask Generation`**: It's crucial to emphasize that the creation of each mask is independent of the others. This independence allows for parallel processing, meaning that the time consumption doesn't increase linearly as one might expect.
>          **(3)** **`In fact, the computation required to generate one mask candidate is far less than one forward and backward pass in BERT`**: Let's delve into the computational cost of generating a single mask candidate to understand our method's efficiency better.
>
>         **3.1** Let's first calculate the required computation by mask candidate generation step by step.
>
>        ```python
>        reshape_weight = weight.cpu().reshape(-1)
>        eshape_weight = reshape_weight ** 5
>        weight_probality = abs(reshape_weight) / sum(abs(reshape_weight))
>
>        masks = torch.zeros_like(reshape_weight)
>        sample_times = max(int((len(reshape_weight) - keep_k) * self._sample_ratio), 1)
>        for i in range(sample_times):
>             sampling_mask = weight_probality.multinomial(
>                 num_samples=keep_k,
>                 replacement=False
>             )
>             masks[sampling_mask] += 1
>             index = torch.topk(masks, keep_k)[1]
>        masks[:] = 0
>        masks[index] = 1
>        mask = masks.reshape(weight.shape).to(weight.device)
>         ```
>
>       1. **Reshaping and Probability Computation**: This process is `4 x d^2` FLOPs for `d x d` weight matrices and `4 x 4d x d` FLOPs for `4d x d` weight matrices.
>
>       2. **Sampling using `multinomial`**: Given the average sparsity from the schedule and `sample_times`, the computation for the `d x d` matrices is approximately `avg_sparsity x avg_sample_times x d^2` FLOPs. For the `4d x d` matrices, it is `avg_sparsity x avg_sample_times x 4d x d` FLOPs.
>
>       3. **Top-k Operation**: For `d x d` matrices, it's about `d^2 x log2(d^2 x avg_sparsity)` FLOPs. For `4d x d` matrices, it is `4d x d x log2(4d x d x avg_sparsity)` FLOPs.
>
>       4. **Total**: Given BERT base's architecture with 12 blocks, each having 4 `d x d` layers and 2 `4d x d` layers, the total computation for pruning the entire BERT model is **1.34 GFLOPs** (with 1.34 being the computed value).
>
>       **3.2** To provide a more intuitive comparison of the computation required, we calculated both the computational cost for one forward pass and one backward pass of Bert.
>
>          1. **Multi-Head Attention**:
>             - Q, K, V calculations: $\(3 \times d \times d \times s\)$
>             - Scaled dot-product attention: $\(d \times s \times d \times s\)$
>             - Output projection: $\(d \times d \times s\)$
>          2. **Feed Forward Network**:
>             - First layer: $\(4d \times d \times s\)$
>             - Second layer: $\(d \times 4d \times s\)$
>          3. **Total**: Given the architecture of BERT base with 12 blocks, the total FLOPs for one forward pass is: $\[ \text{FLOPs for one forward pass} = 12 \times (3d^2s + d^2s^2 + d^2s + 4d^2s + d^2s) \]$. For the backward pass, it's approximately three times the forward pass, equating to **496.47 GFLOPs** for both forward and backward passes combined.
>
>          **3.3** Comparison: Given the above analysis, single mask candidate generation requires **1.34 GFLOPs** for the entire BERT model, whereas a single forward and backward pass of BERT demands **496.7 GFLOPs**. Importantly, while computationally involved, pruning operations are not executed as frequently as the forward and backward passes during training. Hence, over a training session, the added computational cost due to randomized mask generation remains relatively low.
>
>     * **Randomized Mask Selection:**
>         * Requires only **one epoch** to identify the optimal mask.
>         * Consequently, the overhead here is minimal in the grand scheme of the entire pruning process.
>
>   - **Inference Phase**: It's essential to note that when assessing sparse neural networks' deployment in real-world scenarios, the computational effort during the pruning phase is a significant factor. Thus, even if there's a computational overhead during this phase, it's often justified by the benefits obtained during inference in production environments. Moreover, our method not only demonstrates superior performance at a 16x compression rate compared to other methods, but it also maintains performance at even higher sparsity levels, achieving up to a 100x compression rate. This means that when our pruned neural networks are deployed in production settings, the gains in performance and efficiency are even more pronounced.
>
> ### Question 5: How does ir work in your methods?
>
> We appreciate the opportunity to clarify the intricacies of our methodology further.
>
> 1. **`[Line 226-235]` Clarification on the Role of $\( ir \)$**: The equation (Equation 1) quantifies how much our random pruning method diverges from a deterministic approach. Essentially, the $\( ir \)$ value gauges the degree of randomness introduced by our random method relative to the deterministic one. A crucial aspect is that a lower $\( ir \)$ value signifies a closer resemblance to deterministic masking, which may be desirable in some cases, but our method aims to harness the power of certain randomness levels.
>
> 2. **`[Line 238-246]` Handling the Randomness via Ensemble Masks**: While randomness is an integral part of our methodology, it is also important to ensure that it does not spiral out of control. Hence, our ensemble mask approach. By aggregating $\( M \)$ masks and setting top-k values to 1, we can curate a mask with controlled randomness. The $\( M \)$ value directly impacts the magnitude of randomness.
>
> 3. **`[Line408-420]` Hyper-parameter $\( sr \)$ and its Function**: The hyper-parameter $\( sr \)$ offers control over the total introduced randomness by regulating the number of sampled masks $\( M \)$. It is instrumental in tailoring our pruning strategy according to specific requirements, allowing for a dynamic adaptation of the level of randomness.
>
> 4. **`[Line 424-440]` Comparison of Randomness Schedules**: Our experiments aimed to highlight the implications of varying the introduced randomness. As shown in Figure 3, the findings underscore the importance of controlling randomness. Interestingly, our results show that there are indeed regions where our random approach outperforms a deterministic one, reaffirming the significance of our methodology.
>
> 5. **`[Line1088-1093]` Hyperparameter Settings**: We meticulously chose the $\( sr \)$ range based on preliminary experiments to ensure that the randomness remains within the desired bounds. The exponential function parameter $\( T \)$ accelerates the reduction of randomness, providing a balance in the method.
>
> 6. **In conclusion**, the concept of $\( ir \)$ is central to our methodology, as it measures the introduced randomness, but our framework employs multiple strategies to manage and control this randomness for improved pruning performance. We believe that by blending deterministic principles with controlled randomness, we achieve a balanced approach that capitalizes on the strengths of both strategies.
>
> ### Question 6: Do you consider applying the method on larger models in addition to BERT?
>
> Certainly, the idea of scaling our method to larger language models, such as those at the billion parameter level, is intriguing. However, there are challenges to consider. **The current state of pruning for models of this magnitude is still a conundrum**. SparseGPT, for instance, is a recent method introduced for pruning at the billion-parameter level. It's crucial to note that SparseGPT is a post-training pruning method, requiring no further training. Therefore, its effectiveness is mainly evident at a 2x compression level. While this provides a significant benchmark, our approach might not directly apply. Nonetheless, we are actively exploring the feasibility of introducing randomness on the foundation of this method.
>
> **Finally, we thank the valuable suggestions from the reviewer again.**

---

### Official Review · Reviewer_xLoZ · 2023-08-03

**Typos Grammar Style And Presentation Improvements:** 1. Line 133 ‘others than others’
2. P…
**Soundness:** 3

**Excitement:**

3: Ambivalent: It has merits (e.g., it reports state-of-the-art results, the idea is nice), but there are key weaknesses (e.g., it describes incremental work), and it can significantly benefit from another round of revision. However, I won't object to accepting it if my co-reviewers champion it.

**Paper Topic And Main Contributions:**

This article belongs to the lightweight field of NLP model, which is improved on the IMP iterative amplitude pruning. The weight of amplitude pruning near the threshold accounts for about 30%, and simply pruning the weights smaller than the threshold may not be the optimal solution. In view of the above problems, this paper derives the probability distribution by standardizing the absolute value of weights, then randomly samples multiple times, retains the first k weights to obtain masks, and obtains multiple masks repeatedly to select the optimal mask through training with a large learning rate of 1epoch. After experiments, the method proposed in this paper performs better than the original method in all tasks.

**Questions For The Authors:**

1. What is the time and calculation cost of the method proposed in this paper?
2. Distillation is mentioned but not explained in Section 3


**Reasons To Accept:**

1. The description in this article is clear, the pseudo-code given is highly readable, and has an acc increase in the experiments of all tasks.
2. The theoretical argument provided for why this method works is reasonable.
3. The influence of each step in the pipeline is gradually analyzed, and the ablation experiment is complete.


**Reasons To Reject:**

1. The rate of increase is too small, most of which are less than 1%.
2. The amount of calculation is large, and it is difficult to apply to the current large language model with billions of parameters.
3. It is not clear how to find a suitable ir to prune different models.


**Reproducibility:**

4: Could mostly reproduce the results, but there may be some variation because of sample variance or minor variations in their interpretation of the protocol or method.

**Reviewer Confidence:**

3: Pretty sure, but there's a chance I missed something. Although I have a good feel for this area in general, I did not carefully check the paper's details, e.g., the math, experimental design, or novelty.

---

> ### Author Rebuttal · Authors · 2023-08-25
>
> We sincerely thank the reviewer for their insightful comments and questions, which have offered us a clear perspective on areas for improvement. We have carefully gone through the feedback and summarized the major points raised. Below, we have listed the primary concerns and questions from the review to address them systematically in our rebuttal.
>
> Here is a summarized list of the questions extracted from the content:
>
> - **Question 1:** Is the performance gain from your method significant, especially in comparison to other methods like IMP?
> - **Question 2:** Is applying the method to the current large language model with billions of parameters difficult?
> - **Question 3:** How do you find a suitable ir to prune different models?
> - **Question 4:** How efficient is your method?
> - **Question 5:** Distillation is mentioned but not explained in Section 3
> - **Question 6:** Presentation Improvement
>
> Please see our responses below to clarify the main concerns. Also, if you feel that your original concerns have been resolved, we would appreciate it if you would update your evaluation to reflect this. Thank you!
>
> ### Question 1:  Is the performance gain from your method significant, especially in comparison to other methods like IMP?
>
> 1. **Performance Improvement**: In our study, we did observe a modest improvement in the range of 0.2~2.6% at a sparsity level of 16x. Nonetheless, as can be seen from Table I in our paper, our method consistently outperforms all the baselines at similar sparsity levels. The best results for each dataset are highlighted in bold. At a 16x compression rate (with 6% of the parameters retained), the performance advantage of our strategy over IMP for each dataset is as outlined below:
>
>       | Paper  | QNLI | RTE | MRPC | QQP | SST-2 | MNLI | CoLA | STS-B |
>       | ------ | ----: | ----: | ----: | ----: | ----: | ----: | ----: | ----: |
>       | Ours (acc)   | +0.4 | **+2.6** | **+1.3** | +0.3 | **+0.5** | +0.1 | +0.3 | **+0.5** |
>       | Ours (std)   |  0.1 |  0.4 | 0.1 | 0.07 | 0.1 | 0.02 | 0.1 | 0.2 |
>       | Ref    | -0.5 | +1.5 | +1.2 | +0.1 | +0.9 | +1.2 | N/A  | -1.9 |
>
>       For comparison, we also included the performance gains reported on these datasets from a previous study titled `Structured Pruning Learns Compact and Accurate Models` published at `ACL 2022`. Given these results, we believe the gains we achieved are noteworthy.
>
> 2. **Gains at Higher Sparsity Levels**: Our method exhibits even more pronounced improvements, in the range of 2~3%, at higher sparsity levels such as 20x, 40x, 60x, 80x, and 100x, as demonstrated in Figure 5a.
>
> ### Question 2:  Is applying the method to the current large language model with billions of parameters difficult?
>
> 1. **Core Focus:** Our work's central aim is to explore the influence of randomness on traditional deterministic pruning. Direct scalability to LLMs wasn't the primary intent but understanding this interplay is foundational for future research.
>
> 2. **Comparison to Existing Methods:** While methods like SparseGPT are tailored for LLMs, they don't delve into randomness's role in pruning. It's crucial for the community to understand how randomness can influence pruning before we delve deep into scaling up these techniques.
>
> 3. **Evolutionary Nature of Research:** Initial research often starts with foundational exploration, leading to more scalable methods over time. We see our work as an initial step, which, while not immediately scalable, provides insights for future refinements. Recognizing the reviewer's point, we can suggest potential approaches to scale controllable random pruning methods for large language models, ensuring the broader community understands our findings and future directions.
>
> ### Question 3: How do you find a suitable ir to prune different models?
>
> We've elaborated on this in our paper, and specific locations of these details are provided below.
>
> 1. **`[Line 226-235]` Firstly, the parameter `$ir$` is introduced as a metric to quantify the amount of randomness we insert during the pruning process.** As described by Equation 1, it measures the deviation of our random approach from a deterministic one. In simple terms, `$ir$` is merely an evaluative measure of the randomness we introduce, not a direct control parameter for pruning different models.
>
> 2. **`[Line 238-246]` The actual control of how much randomness to introduce is governed by the number of masks we sample, denoted as `$M$`.** This `$M$` is directly influenced by the sample ratio, `$sr$`, which serves as our hyper-parameter. As detailed in our manuscript, the more masks we sample (higher `$M$`), the greater the randomness introduced. Conversely, sampling fewer masks will lead to pruning closer to deterministic methods.
>
> 3. **`[Section 4.5.1: Impact of Randomness and Schedule]` We have exhaustively explored the influence of varying levels of introduced randomness (via `$ir$`) on the pruning results in our ablation study.** Figure 3 shows how different schedules of introducing randomness impact the pruning efficiency. It’s worth noting that an optimal threshold of randomness exists below which our method consistently outperforms deterministic pruning (IMP). These findings emphasize the advantage of introducing controlled randomness over purely deterministic pruning strategies.
>
> 4. **`[Line 1088-1090]` Further, in the appendix of our manuscript, we have detailed the hyper-parameters (especially the sample ratio, `$sr$`) used across different experiments**, offering guidance on their applicability to various models. Our experiments with different `$sr$` values, such as `3e-6`, `5e-6`, and so forth, can serve as a reference for practitioners.
>
> 5. **To conclude, while `$ir$` serves as a measure of the introduced randomness, the actual control over the randomness is exerted through the number of sampled masks `$M$`, governed by the sample ratio `$sr$`.** We believe our thorough experiments and discussions in the main manuscript and appendix address the concerns raised.
>
> ### Question 4:  How efficient is your method?
>
> In response to your query about the efficiency of our method, we address this concern from two perspectives: the training and inference phases.
>
>   - **Training Phase**: Concerning the computational overhead of our method
>     * **Randomized Mask Generation:**
>          **(1)** At first glance, the computation in this phase might seem directly proportional to the number of randomized mask candidates we aim to generate.
>          **(2)** **`Utilizing Parallel Processing to Expedite Randomized Mask Generation`**: It's crucial to emphasize that the creation of each mask is independent of the others. This independence allows for parallel processing, meaning that the time consumption doesn't increase linearly as one might expect.
>          **(3)** **`In fact, the computation required to generate one mask candidate is far less than one forward and backward pass in BERT`**: Let's delve into the computational cost of generating a single mask candidate to understand our method's efficiency better.
>
>         **3.1** Let's first calculate the required computation by mask candidate generation step by step.
>
>        ```python
>        reshape_weight = weight.cpu().reshape(-1)
>        eshape_weight = reshape_weight ** 5
>        weight_probality = abs(reshape_weight) / sum(abs(reshape_weight))
>
>        masks = torch.zeros_like(reshape_weight)
>        sample_times = max(int((len(reshape_weight) - keep_k) * self._sample_ratio), 1)
>        for i in range(sample_times):
>             sampling_mask = weight_probality.multinomial(
>                 num_samples=keep_k,
>                 replacement=False
>             )
>             masks[sampling_mask] += 1
>             index = torch.topk(masks, keep_k)[1]
>        masks[:] = 0
>        masks[index] = 1
>        mask = masks.reshape(weight.shape).to(weight.device)
>         ```
>
>       1. **Reshaping and Probability Computation**: This process is `4 x d^2` FLOPs for `d x d` weight matrices and `4 x 4d x d` FLOPs for `4d x d` weight matrices.
>
>       2. **Sampling using `multinomial`**: Given the average sparsity from the schedule and `sample_times`, the computation for the `d x d` matrices is approximately `avg_sparsity x avg_sample_times x d^2` FLOPs. For the `4d x d` matrices, it is `avg_sparsity x avg_sample_times x 4d x d` FLOPs.
>
>       3. **Top-k Operation**: For `d x d` matrices, it's about `d^2 x log2(d^2 x avg_sparsity)` FLOPs. For `4d x d` matrices, it is `4d x d x log2(4d x d x avg_sparsity)` FLOPs.
>
>       4. **Total**: Given BERT base's architecture with 12 blocks, each having 4 `d x d` layers and 2 `4d x d` layers, the total computation for pruning the entire BERT model is **1.34 GFLOPs** (with 1.34 being the computed value).
>
>       **3.2** To provide a more intuitive comparison of the computation required, we calculated both the computational cost for one forward pass and one backward pass of Bert.
>
>          1. **Multi-Head Attention**:
>             - Q, K, V calculations: $\(3 \times d \times d \times s\)$
>             - Scaled dot-product attention: $\(d \times s \times d \times s\)$
>             - Output projection: $\(d \times d \times s\)$
>          2. **Feed Forward Network**:
>             - First layer: $\(4d \times d \times s\)$
>             - Second layer: $\(d \times 4d \times s\)$
>          3. **Total**: Given the architecture of BERT base with 12 blocks, the total FLOPs for one forward pass is: $\[ \text{FLOPs for one forward pass} = 12 \times (3d^2s + d^2s^2 + d^2s + 4d^2s + d^2s) \]$. For the backward pass, it's approximately three times the forward pass, equating to **496.47 GFLOPs** for both forward and backward passes combined.
>
>          **3.3** Comparison: Given the above analysis, single mask candidate generation requires **1.34 GFLOPs** for the entire BERT model, whereas a single forward and backward pass of BERT demands **496.7 GFLOPs**. Importantly, while computationally involved, pruning operations are not executed as frequently as the forward and backward passes during training. Hence, over a training session, the added computational cost due to randomized mask generation remains relatively low.
>
>     * **Randomized Mask Selection:**
>         * Requires only **one epoch** to identify the optimal mask.
>         * Consequently, the overhead here is minimal in the grand scheme of the entire pruning process.
>
>   - **Inference Phase**: It's essential to note that when assessing sparse neural networks' deployment in real-world scenarios, the computational effort during the pruning phase is a significant factor. Thus, even if there's a computational overhead during this phase, it's often justified by the benefits obtained during inference in production environments. Moreover, our method not only demonstrates superior performance at a 16x compression rate compared to other methods, but it also maintains performance at even higher sparsity levels, achieving up to a 100x compression rate. This means that when our pruned neural networks are deployed in production settings, the gains in performance and efficiency are even more pronounced.
>
> ### Question 5: Distillation is mentioned but not explained in Section 3
>
> 1. **Placement in Preliminaries**:  In the Preliminaries, we indeed touched upon the concept of Knowledge Distillation (KD) as an essential background. We intended to provide a foundational understanding of this technique as it is used in the literature.
>
> 2. **Relation to Our Methodology**:  While Knowledge Distillation was not detailed in the Methodology section, it plays a pivotal role in our experimental setup. Specifically, in the Setup subsection, it is clearly highlighted that we incorporate KD in our approach. By stating, "Our experiments evaluate the pruning methods based on BERT~\citep{Devlin:2019}, and we apply the KD method to both the baseline and our strategy," we imply that KD is an integral part of the experimentation.
>
> 3. **Connection to the Baseline**:  It is crucial to emphasize that our primary baseline, IMP, and our proposed strategy both utilize the same distillation method. This methodological consistency ensures that any performance differences observed between our proposed strategy and the baseline can be attributed to our novel contributions rather than variations in the distillation approach.
>
> 4. **Knowledge Gained from Literature**:  It is important to highlight that our decision to use KD (Knowledge Distillation) to enhance performance was influenced by the work presented in SparseBert~\citep{Xu:2021}. Our goal was to employ a proven technique to attain the best results. Additionally, many pruning methods in NLP now consider knowledge distillation a standard practice.
>
> 5. **Decision on Exclusion from Methodology**:  The reason for not explicating KD in the Methodology section was our judgment that it wasn't the focal point of our work. Since KD is a widely accepted technique, we believe the emphasis should be on our unique contributions. Nevertheless, we acknowledge the importance of ensuring clarity and will consider providing a brief note on its role within the methodology in future versions.
>
> 6. **In-Depth Explanation Provided**:  For readers interested in understanding our specific application of KD, we've included a dedicated subsection on Knowledge Distillation, elucidating how we distill knowledge from the hidden states and attention scores. This ensures that those keen on the details have access to this information.
>
> 7. **In summary**: while Knowledge Distillation was not explicitly discussed in the methodology, it is woven into the fabric of our experimental setup. We appreciate the feedback and will ensure that all pivotal components, even if they're not the primary focus, are adequately explained in future work to avoid ambiguity.
>
> ### Question 6: Presentation Improvement
>
> Thank you for your meticulous observations regarding the typos and presentation in our manuscript. We sincerely appreciate your time and effort to provide this feedback.
>
> 1. We have noted the error on Line 133 "others than others" and will rectify it in the revised version.
> 2. The double semicolons in Pseudocode 3 and 4 will be addressed and corrected.
> 3. We will clarify and correct the discrepancy where "k is equal y" in the pseudocode comments.
> 4. The typo on Line 252 "of conresponding" will also be rectified.
>
> Your feedback has been instrumental in enhancing the clarity and accuracy of our work. We will ensure that these points are addressed in our revised version.
>
> **Finally, we thank the valuable suggestions from the reviewer again.**

---

### Official Review · Reviewer_cTJA · 2023-08-05

**Soundness:** 4

**Excitement:**

3: Ambivalent: It has merits (e.g., it reports state-of-the-art results, the idea is nice), but there are key weaknesses (e.g., it describes incremental work), and it can significantly benefit from another round of revision. However, I won't object to accepting it if my co-reviewers champion it.

**Paper Topic And Main Contributions:**

the paper proposed that the deterministic-based method for pruning is sub-optimal and they proposed a pruning method that has controllable randomness for pruning the search space which could lead to better performance

**Questions For The Authors:**

- could you please elaborate a little more on the motivation? especially the problem with magnitude-based methods. please give some clarification on the lack of variety and the example in Figure 1. Why τ = 0.027 and τ = 0.055 set and why 29% cannot be simply ignored even if these 29 percent of params can be pruned based on the magnitude?
-regarding question 2 in page 2: (Can we design a consistently effective randomized pruning method) I'm not sure how efficient your method is compared with a method like IMP? did you compare these two not just for performance but how efficient they are?

- although the proposed method beats existing methods compared with IMP that you implemented yourself, the difference is not significant ~0.1-0.5%. do you know which one is better in other aspects like computation? IMP or your method?

**Reasons To Accept:**

achieving SOTA in 8 tasks of GLUE.
proposed the idea of controllable randomness is interesting
extensive experiments prove the idea works (although the margin is not significant)

**Reasons To Reject:**

- motivation for proposing an alternative approach for controllable randomness and the advantage over magnitude-based method is missing
- the results of the proposed method beat SOTA. However, the difference is not significant. additionally the analysis of how the proposed framework is efficient compared with other methods like IMP is missing

**Reproducibility:**

3: Could reproduce the results with some difficulty. The settings of parameters are underspecified or subjectively determined; the training/evaluation data are not widely available.

**Reviewer Confidence:**

3: Pretty sure, but there's a chance I missed something. Although I have a good feel for this area in general, I did not carefully check the paper's details, e.g., the math, experimental design, or novelty.

---

> ### Author Rebuttal · Authors · 2023-08-25
>
> We sincerely thank the reviewer for their insightful comments and questions, which have offered us a clear perspective on areas for improvement. We have carefully gone through the feedback and summarized the major points raised. Below, we have listed the primary concerns and questions from the review to address them systematically in our rebuttal.
>
> Here is a summarized list of the questions extracted from the content:
>
> ### Motivation Related:
> - **Question 1:** What is the limitation of deterministic magnitude-based pruning methods?
>   - Concerns about:
>     - Lack of variety in the approach.
>
> - **Question 2:** Can you explain Figure 1, specifically regarding:
>   - The reasoning behind selecting specific values of $\( \tau \)$ (i.e., $\( \tau = 0.027 \) and \( \tau = 0.055 \)$).
>   - The significance of the 29% in the range $\( \left[ \frac{2}{3} \tau, \frac{4}{3} \tau \right] \)$.
>
> ### Performance Gain and Pruning Cost Related:
> - **Question 3:** Is the performance gain from your method significant, especially in comparison to other methods like IMP?
> - **Question 4:** How efficient is your method?
>
> Please see our responses below to clarify the main concerns. Also, if you feel that your original concerns have been resolved, we would appreciate it if you would update your evaluation to reflect this. Thank you!
>
> ## Motivation Related
>
> As highlighted in the introduction (`Line 086~091, 103~108`), traditional deterministic pruning methods are constrained by their sole reliance on magnitude information, resulting in limited adaptability. In contrast, this paper proposes an approach that infuses controlled randomness into the pruning strategy, striving to find a middle ground between fully deterministic and entirely random methods.
>
> ---
> ### Question 1. What is the limitation of deterministic magnitude-based pruning methods?
>
> Deterministic magnitude-based pruning methods aim to trim weights, neurons, or layers from a neural network that appear least influential. The absolute value of their weights primarily determines this. The rationale behind these methods is that eliminating what seem to be the least consequential elements can reduce the overall model size and computational requirements without a drastic loss in performance.
>
> However, several challenges arise with this approach:
>
> 1. **Lack of Variety:** Operating strictly under a deterministic paradigm, these methods focus solely on the magnitude of the weights for pruning decisions. Consequently, they might miss out on smaller weights that, despite their size, play pivotal roles in certain tasks, especially when handling edge cases or rarer instances.
>
>    **Let's consider the following small case:**
>
>    - Neural Network Output:
>       $\[ Y = w_i \times F_i \]$
>
>     **Where:**
>     - $\( Y \)$ is the network output.
>     - $\( F_i \)$ represents a feature.
>     - $\( w_i \)$ is the corresponding weight.
>
>    **Magnitude-based Pruning:**
>     - If $\( |w_i| < T \)$, then $\( w_i \times F_i \)$ is pruned.
>
>    **The Highlighted Risk:**
>     - The impact on $\( Y \)$ due to a feature is not solely determined by $\( w_i \)$. Instead, it's the combined effect of $\( w_i \)$ and the sensitivity of $\( F_i \)$.
>       - **High sensitivity (Example)**: Suppose $\( F_i \)$ represents the sharpness of an image, a small change in sharpness (even if $\( w_i \)$ is small) might drastically affect our recognition of an object in the image. Thus, even if $\( |w_i| = 0.01 \)$ (small weight), the influence of $\( F_i \)$ could make a difference of 0.5 in $\( Y \)$, making it critical.
>
>       - **Low sensitivity (Example)**: On the other hand, consider $\( F_i \)$ representing the hue of the image background. Even if this hue has a large weight $\( w_i = 5 \)$, a change in the background hue might not alter our recognition much, making the product $\( w_i \cdot F_i \)$ yield only a minimal difference of 0.1 in $\( Y \)$.
>
>    **Conclusion:**
>     - The influence of removing weights on $\( Y \)$ is not solely reliant on $\( w_i \)$. Instead, it's the joint effect of $\( w_i \)$ and $\( F_i \)$'s sensitivity. Hence, the deterministic nature might lead to a "lack of variety," potentially overlooking these nuanced relationships.
>
> 2. **Sub-optimal at high-level sparsity:**
>    It's essential to underscore that deterministic magnitude-based pruning exhibits suboptimal performance when pushed to extreme sparsity levels. This limitation becomes evident at high compression rates such as 20x, 40x, 60x, 80x and 100x. Our exhaustive experiments on the BERT model across datasets like MRPC, RPC, and MNLI corroborate this observation. Notably, utilizing the Iterative Magnitude Pruning (IMP) techniques largely deteriorated performance at these higher sparsity levels. The table below details the performance degradation for various datasets and sparsity levels when using deterministic and our randomized iterative magnitude-based pruning:
>
>    | Method (IMP) | Dataset | 20x Compression  | 40x Compression| 60x Compression | 80x Compression | 100x Compression |
>    |---------|---------|-----------------|-----------------|-----------------|-----------------|------------------|
>    | Determinstic | MRPC    |  -2.6% |  -2.7% | -5.0% |  -6.1%   | -6.7%   |
>    | **Randomized** | MRPC    | **-0.5%** | **-1.4%** | **-2.9%** | **-3.7%** | **-3.9%** |
>    | Determinstic | RPC     |  -2.8% |  -3.5% | -5.7% |  -7.4%   | -8.3%   |
>    | **Randomized** | RPC    |  **-0.2%** |  **-1.1%** | **-2.3%** | **-3.5%**  | **-4.0%**  |
>
> ### Question 2. Can you explain Figure 1?
>
>   1. **Weight Distribution of BERT Layer:**
> The figure displays the weight distribution for a particular BERT layer, spotlighting weights situated around two specific pruning thresholds: $\( \tau = 0.027 \)$ and $\( \tau = 0.055 \)$.
>
>   2. **Selection of $\( \tau = 0.027 \) and \( \tau = 0.055 \)$:**
> The selected $\( \tau \)$ values directly relate to sparsity levels of 0.52 and 0.83. Their selection is intricately connected to the Iterative Magnitude Pruning (IMP) process. Within specific phases of IMP, the task necessitates pruning 52% or 83% of the weights, guided by their magnitude. These $\( \tau \)$ values act as exact pruning thresholds. It's pivotal to understand that these figures aren't arbitrary. Their determination is primarily anchored in the established IMP pruning schedule. In our illustration, while we highlighted sparsity levels of 0.52 and 0.83, it's noteworthy that analogous outcomes were consistently observed across various sparsity levels.
>
>   3. **The Significance of the 29% in the Range $\([ \frac{2}{3} \tau, \frac{4}{3} \tau ]\)$:**
> I apologize for any earlier ambiguities. While magnitude is undeniably significant, it doesn't singularly determine performance. Even weights with smaller magnitudes can be crucial, especially when dealing with edge cases or infrequent situations. Proximity between weights can intensify decision-making challenges. Importantly, weights within this particular range, which approach the pruning threshold, constitute 29% of all weights. This notable fraction emphasizes their potential outsized influence on the total output, marking their indispensable role.
>
> ## Performance Gain and Pruning Cost Related
> ---
>
> Thank you for pointing out the comparison between our method and IMP. To clarify:
>
> ### Question 3:  Performance Gain is not Significant
>
> 1. **Performance Improvement**: In our study, we did observe a modest improvement in the range of 0.1~2.6% at a sparsity level of 16x. Nonetheless, as can be seen from Table I in our paper, our method consistently outperforms all the baselines at similar sparsity levels. The best results for each dataset are highlighted in bold. At a 16x compression rate (with 6% of the parameters retained), the performance advantage of our strategy over IMP for each dataset is as outlined below:
>
>       | Paper  | QNLI | RTE | MRPC | QQP | SST-2 | MNLI | CoLA | STS-B |
>       | ------ | ----: | ----: | ----: | ----: | ----: | ----: | ----: | ----: |
>       | Ours (acc)   | +0.4 | **+2.6** | **+1.3** | +0.3 | **+0.5** | +0.1 | +0.3 | **+0.5** |
>       | Ours (std)   |  0.1 |  0.4 | 0.1 | 0.07 | 0.1 | 0.02 | 0.1 | 0.2 |
>       | Ref    | -0.5 | +1.5 | +1.2 | +0.1 | +0.9 | +1.2 | N/A  | -1.9 |
>
>       For comparison, we also included the performance gains reported on these datasets from a previous study titled `Structured Pruning Learns Compact and Accurate Models` published at `ACL 2022`. Given these results, we believe the gains we achieved are noteworthy.
>
> 2. **Gains at Higher Sparsity Levels**: Our method exhibits even more pronounced improvements, in the range of 2~4%, at higher sparsity levels such as 20x, 40x, 60x, 80x, and 100x, as demonstrated in Figure 5a.
>
> ### Question 4:  How efficient is your method?
>  In response to your query about the efficiency of our method, we address this concern from two perspectives: the training and inference phases.
>
>   - **Training Phase**: Concerning the computational overhead of our method
>     * **Randomized Mask Generation:**
>          **(1)** At first glance, the computation in this phase might seem directly proportional to the number of randomized mask candidates we aim to generate.
>          **(2)** **`Utilizing Parallel Processing to Expedite Randomized Mask Generation`**: It's crucial to emphasize that the creation of each mask is independent of the others. This independence allows for parallel processing, meaning that the time consumption doesn't increase linearly as one might expect.
>          **(3)** **`In fact, the computation required to generate one mask candidate is far less than one forward and backward pass in BERT`**: Let's delve into the computational cost of generating a single mask candidate to understand our method's efficiency better.
>
>         **3.1** Let's first calculate the required computation by mask candidate generation step by step.
>
>        ```python
>        reshape_weight = weight.cpu().reshape(-1)
>        eshape_weight = reshape_weight ** 5
>        weight_probality = abs(reshape_weight) / sum(abs(reshape_weight))
>
>        masks = torch.zeros_like(reshape_weight)
>        sample_times = max(int((len(reshape_weight) - keep_k) * self._sample_ratio), 1)
>        for i in range(sample_times):
>             sampling_mask = weight_probality.multinomial(
>                 num_samples=keep_k,
>                 replacement=False
>             )
>             masks[sampling_mask] += 1
>             index = torch.topk(masks, keep_k)[1]
>        masks[:] = 0
>        masks[index] = 1
>        mask = masks.reshape(weight.shape).to(weight.device)
>         ```
>
>       1. **Reshaping and Probability Computation**: This process is `4 x d^2` FLOPs for `d x d` weight matrices and `4 x 4d x d` FLOPs for `4d x d` weight matrices.
>
>       2. **Sampling using `multinomial`**: Given the average sparsity from the schedule and `sample_times`, the computation for the `d x d` matrices is approximately `avg_sparsity x avg_sample_times x d^2` FLOPs. For the `4d x d` matrices, it is `avg_sparsity x avg_sample_times x 4d x d` FLOPs.
>
>       3. **Top-k Operation**: For `d x d` matrices, it's about `d^2 x log2(d^2 x avg_sparsity)` FLOPs. For `4d x d` matrices, it is `4d x d x log2(4d x d x avg_sparsity)` FLOPs.
>
>       4. **Total**: Given BERT base's architecture with 12 blocks, each having 4 `d x d` layers and 2 `4d x d` layers, the total computation for pruning the entire BERT model is **1.34 GFLOPs** (with 1.34 being the computed value).
>
>       **3.2** To provide a more intuitive comparison of the computation required, we calculated both the computational cost for one forward pass and one backward pass of Bert.
>
>          1. **Multi-Head Attention**:
>             - Q, K, V calculations: $\(3 \times d \times d \times s\)$
>             - Scaled dot-product attention: $\(d \times s \times d \times s\)$
>             - Output projection: $\(d \times d \times s\)$
>          2. **Feed Forward Network**:
>             - First layer: $\(4d \times d \times s\)$
>             - Second layer: $\(d \times 4d \times s\)$
>          3. **Total**: Given the architecture of BERT base with 12 blocks, the total FLOPs for one forward pass is: $\[ \text{FLOPs for one forward pass} = 12 \times (3d^2s + d^2s^2 + d^2s + 4d^2s + d^2s) \]$. For the backward pass, it's approximately three times the forward pass, equating to **496.47 GFLOPs** for both forward and backward passes combined.
>
>          **3.3** Comparison: Given the above analysis, single mask candidate generation requires **1.34 GFLOPs** for the entire BERT model, whereas a single forward and backward pass of BERT demands **496.7 GFLOPs**. Importantly, while computationally involved, pruning operations are not executed as frequently as the forward and backward passes during training. Hence, over a training session, the added computational cost due to randomized mask generation remains relatively low.
>
>     * **Randomized Mask Selection:**
>         * Requires only **one epoch** to identify the optimal mask.
>         * Consequently, the overhead here is minimal in the grand scheme of the entire pruning process.
>
>   - **Inference Phase**: It's essential to note that when assessing sparse neural networks' deployment in real-world scenarios, the computational effort during the pruning phase is a significant factor. Thus, even if there's a computational overhead during this phase, it's often justified by the benefits obtained during inference in production environments. Moreover, our method not only demonstrates superior performance at a 16x compression rate compared to other methods, but it also maintains performance at even higher sparsity levels, achieving up to a 100x compression rate. This means that when our pruned neural networks are deployed in production settings, the gains in performance and efficiency are even more pronounced.
>
> **Finally, we thank the valuable suggestions from the reviewer again.**

---

### Meta-Review · Area_Chair_9Uce · 2023-09-18

**Recommendation:** 3

**Metareview:**

This paper proposes a randomized pruning approach that is adapted from iterative magnitude pruning (IMP), where multiple masks are generated using weight magnitude as part of random selection and then independently evaluated for one epoch. The mask with the best validation score after that epoch is then selected.

Generally, reviewers liked the idea and agree that it improves performance when compared against a commonly used approach (IMP) across many evaluated settings. The method is also quite clear and the addition of pseudo-code for the approach makes it easy to implement.

One point raised by reviewers was the training overhead introduced by this mask selection process. In the rebuttal, the authors provided evidence (via code and floating-point op estimates) to demonstrate that these masks can be generated relatively efficiently (significantly less than one forward/backward pass of the evaluated model. However, the authors do not provide an estimate of the actual training cost of the approach (not just the mask generation). As the paper says that each mask is trained for one epoch (i.e., many forward and backward passes), it seems that this may lead to significant overhead depending on the number of masks per layer, number of layers, and number of pruning stages. A detailed analytical model of this cost and/or training time comparison of this approach compared to IMP would make it more clear how important of an issue this is.

---

### Decision · Program_Chairs · 2023-10-07

**Decision:**

Accept-Findings

**Comment:**

This paper proposes a randomized pruning approach that is adapted from iterative magnitude pruning (IMP), where multiple masks are generated using weight magnitude as part of random selection and then independently evaluated for one epoch. The mask with the best validation score after that epoch is then selected.

Generally, reviewers liked the idea and agree that it improves performance when compared against a commonly used approach (IMP) across many evaluated settings. The method is also quite clear and the addition of pseudo-code for the approach makes it easy to implement.

One point raised by reviewers was the training overhead introduced by this mask selection process. In the rebuttal, the authors provided evidence (via code and floating-point op estimates) to demonstrate that these masks can be generated relatively efficiently (significantly less than one forward/backward pass of the evaluated model. However, the authors do not provide an estimate of the actual training cost of the approach (not just the mask generation). As the paper says that each mask is trained for one epoch (i.e., many forward and backward passes), it seems that this may lead to significant overhead depending on the number of masks per layer, number of layers, and number of pruning stages. A detailed analytical model of this cost and/or training time comparison of this approach compared to IMP would make it more clear how important of an issue this is.